# Cross-Lingual Named Entity Recognition Based on Attention and Adversarial Training

Hao Wang [1,2], Lekai Zhou [1,2], Jianyong Duan [1,2,*] and Li He [1,2]

1   School of Information Science and Technology, North China University of Technology, Beijing 100144, China
2   CNONIX National Standard Application and Promotion Lab, Beijing 100144, China
*   Correspondence: duanjy@ncut.edu.cn

**Abstract:** Named entity recognition aims to extract entities with specific meaning from unstructured text. Currently, deep learning methods have been widely used for this task and have achieved remarkable results, but it is often difficult to achieve better results with less labeled data. To address this problem, this paper proposes a method for cross-lingual entity recognition based on an attention mechanism and adversarial training, using resource-rich language annotation data to migrate to low-resource languages for named entity recognition tasks and outputting changing semantic vectors through the attention mechanism to effectively solve the long-sequence semantic dilution problem. To verify the effectiveness of the proposed method, the method in this paper is applied to the English–Chinese cross-lingual named entity recognition task based on the WeiboNER data set and the People-Daily2004 data set. The obtained F1 value of the optimal model is 53.22% (a 6.29% improvement compared to the baseline). The experimental results show that the cross-lingual adversarial named entity recognition method proposed in this paper can significantly improve the results of named entity recognition in low resource languages.

**Keywords:** named entity recognition; cross-lingual; adversarial training





## 1. Introduction

Cross-lingual name entity recognition is an important but challenging task. An effective solution is to perform cross-lingual transfer by leveraging the annotations from high-resource languages. Most of these efforts achieve cross-lingual annotation projection based on bilingual parallel corpora combined with automatic word alignment and cross-lingual word embedding; however, such resources are still only available for dozens of languages. Named entities are very critical sources of underlying semantic information in text processing tasks, and named entity recognition techniques can be used not only for information extraction, but also for tasks such as automatic text summary, automatic machine answering, machine translation, knowledge graph construction, and machine reading comprehension [1]. In recent years, deep learning methods based on neural network models have been mainly used for named entity recognition tasks, and significant results have been achieved. However, for neural network-based methods, their effectiveness depends heavily on a large amount of manually labeled data, but manually labeling training data is so time-consuming and labor-intensive that the size of the existing corpus for named entity recognition is very limited. Moreover, in some resource-starved languages, manually labeled data do not even exist. The problem of the lack of an annotated corpus exists not only in the research of named entity recognition, but in fact, it is an urgent problem in the whole field of natural language processing research. The significance of research on constructing cross-lingual named entity recognition is that, due to the uneven distribution of knowledge among languages, the shortage of target language data sets can be effectively compensated by data set extension, and entity relationship extraction in low-resource languages can be realized by data set extension; the complementary nature

of multilingualism in knowledge expression can be fully utilized to increase the coverage and sharing of degrees of knowledge [2]. Cross-lingual entity recognition can be applied to tasks such as cross-lingual information extraction, relationship extraction, etc. Due to its wide application prospects, cross-lingual named entity recognition is gaining increasing attention from academia and industry.

In order to solve the above problems and improve the efficiency and accuracy of cross-lingual transfer, this paper combines cross-lingual word vector alignment with cross-lingual named entity recognition tasks, and proposes a named entity recognition framework based on confrontation training in cross-lingual situations. First, the source language and the target language are converted into word vectors through M-BERT [3], and then the self-attention mechanism is reversely fused to convert the information extracted from the English pre-training named entity recognition model into Chinese, strengthen the representation of key information, use the source language to fine-tune the NER model, and use the label data of the source language and the target language to conduct adversarial training, so as to improve the alignment ability of the model between the two languages. Then, the model is used to infer the target language to test the effect of the model on improving the entity recognition in the target language [4]. The experiment shows that the proposed model has achieved good results on WeiboNER and People-Daily2004 data sets, especially on People-Daily2004 data sets with large data volume.

## 2. Related Work

### 2.1. Cross-Lingual Migration

With the development of machine translation research and the emergence of large-scale cross-lingual resources, cross-lingual migration-based approaches enable processing tasks on resource-scarce languages to be solved. The approach of natural language processing based on cross-lingual migration is to migrate the manually annotated data or knowledge from the resource-rich source language with sufficient annotation data to the target language with scarce corpus resources in order to overcome the challenges of languages with little or no annotation data on the target language [5]. For cross-language, migration-based natural language processing research, how to more effectively migrate data or knowledge from high-resource languages with rich entity labels to low-resource target languages is a central issue in this research area.

In the research of cross-lingual named entity recognition, the cross-lingual transfer method based on external cross-lingual resources has also played an important role. Mayhew et al. [6] proposed a cross-lingual named entity recognition method based on bilingual dictionary translation. Xie et al. [7] proposed a method based on bilingual word vectors to realize the translation from source language label data to target language label data. In addition, in order to solve the order problem caused by word-to-word translation, they introduced a self-attention mechanism under the existing neural network architecture. Ni et al. [8] constructed a high-quality multilingual Wikipedia entity type map using weakly labeled data and used the map to improve the performance of the multilingual named entity recognition model. Maud et al. [9] proposed a method for automatically creating multilingual named entity annotation corpus based on a parallel corpus. Pan et al. [10] developed a simple and effective cross-lingual sequence annotation framework. According to the existing research results, the core task of cross-lingual transfer research is to transfer high-quality, manually labeled data or knowledge from resource-rich languages to natural language processing tasks in resource-poor languages.

### 2.2. Adversarial Training

In 2014, Goodfellow et al. [11] proposed the generative adversarial network [12], an unsupervised generative model that has received much attention and research for its powerful data generation capabilities.

The generative adversarial network is not a single network. It has two different networks, one is a generator and the other is a discriminator [13]. The generator takes random

noise as input and the fake samples are generated as the output. The discriminator's purpose is to distinguish the generated fake samples from the real ones. The training of the generative adversarial network is only conducted using the adversarial game approach, and the gradient update information of the generator comes from the discriminator, not from the data. The loss function L (D, G) of the generating adversarial network satisfies [14].

In the formula below, *G* is the generator function, *D* is the discriminator function, *z* is the random noise, *x* is the true sample, $G(z)$ is the generated false sample, $P_{data}(x)$ is the distribution of the true sample, and $P_z(z)$ is the distribution of the generated false sample. The generator and the discriminator are trained alternately; the generator wants to generate more realistic false samples, and the discriminator wants to distinguish the true samples from the false samples as much as possible, so as to play off each other and reach the Nash equilibrium point. In the end, the generator can generate false samples that are seen as true, that is, the discriminator cannot distinguish between true and false samples.

$$\min_{G}\max_{D}L(D,G) = E_{x\sim P_{data}(x)}[lgD(x)] + E_{z\sim P_z(z)}[\lg(1-D(G(z)))] \tag{1}$$

### 2.3. Attention Mechanism

The mechanism of attention originated in the field of biology [15], from the study of human vision. In cognitive science, due to the bottleneck in information processing, humans will selectively pay attention to some parts of all information while ignoring other visible information. These mechanisms are often referred to as attention mechanisms. An informal term for an attention mechanism is that a neural attention mechanism gives a neural network the ability to focus on a subset of its inputs (or features), thus choosing specific inputs. Attention can be applied to any type of input regardless of its shape. In the case of limited computing power, an attention mechanism is a resource allocation scheme that is the main means to solve the problem of information overload, allocating computing resources to more important tasks.

The introduction of attention mechanisms has greatly contributed to research in the field of natural language processing. The attention mechanism was first used in the image domain to highlight the importance of a certain part of an image [16]. The attention mechanism was later introduced in machine translation tasks, which was the first use of an attention mechanism in a model in the field of natural language processing. Since then, many models have used the inclusion of attention mechanisms and various improved versions of attention mechanisms, mostly with very good results. Gradually, the attention mechanism has become a very important technique that has been flexibly used in machine translation, machine reading comprehension, sentiment analysis, named entity recognition, and other natural language processing tasks.

The design of the attention mechanism is inspired by the human visual attention mechanism. When a human observes a picture, they first quickly skim the whole image, then target the area of interest and further devote more attention to that part to obtain more detailed information about it. During the processing of tasks in the field of natural language processing, such as machine reading comprehension, researchers have used the attention mechanism to obtain the dependency between each word of the source sequence and each word of the target sequence, so as to achieve the purpose of having the machine focus on a certain part of the input sequence [17].

During the specific operation using the attention mechanism, the model calculates the weighted average of all elements of the input sequence, which is then fed into the neural network structure after the model for other operations. The calculation process is shown in the figure.

From Figure 1, we can see that the attention value is computed from the three tensors query, key and value. We can treat each item in the source sequence as a key–value pair consisting of a series of <key, value> pairs and an item in the target sequence as query element. By calculating the similarity between the query and each key, we can obtain the

attention weight corresponding to each value and then weigh and sum the weights with the value to finally obtain the attention value [18].

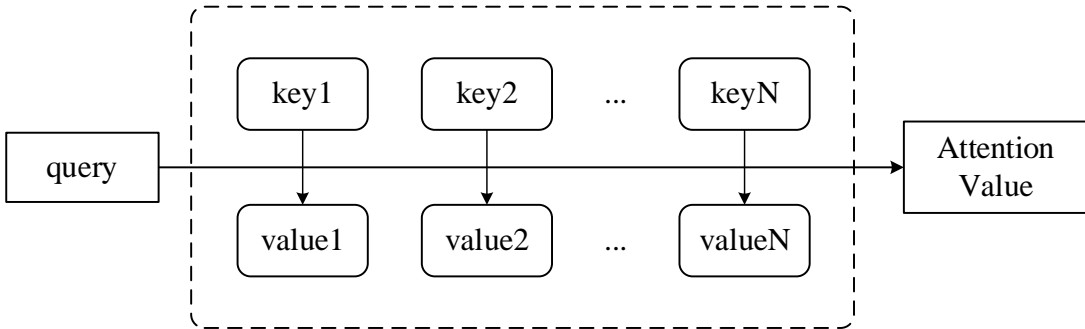

**Figure 1.** Diagram of the calculation process of attention mechanism.

### 3. Model

In order to improve the cross-lingual migration capability of the model, this paper proposes a cross-lingual named entity recognition model based on an attention mechanism and adversarial training. The sequence representation is first obtained through the encoding layer, and the key information in the text is extracted using the attention mechanism. Then, the bilingual word vector is further aligned through the adversarial training layer and passed through the CRF to the output layer. Finally, the labels are output.

The model in this paper contains an embedding layer, a double attention layer, an adversarial training layer, and an output layer, and its structure is shown in Figure 2. The embedding layer consist of M-Bert and source word embedding and target word embedding. The double attention layer consist of LSTM and attention. The adversarial training layer consist of a Word discriminator that will calculate discriminator loss and generator loss. Finally, the CRF layer will output all results.

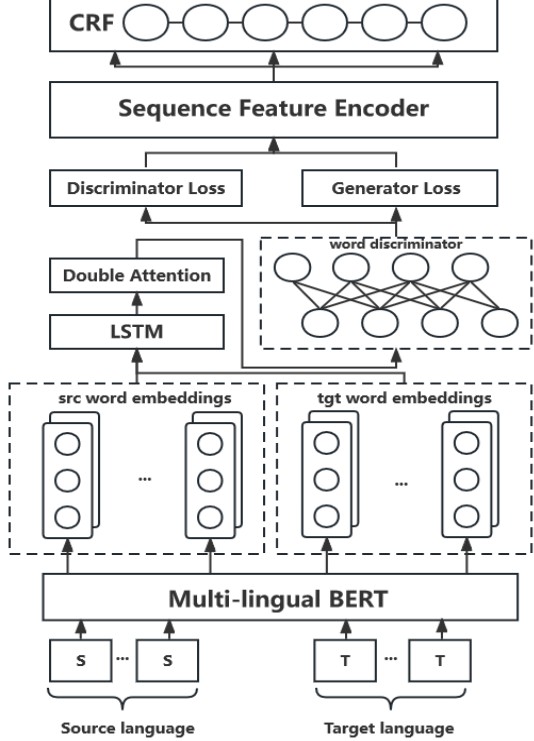

**Figure 2.** Model Structure.

### *3.1. Embedding*

BERT passes each word (token) in the input text through an embedding layer in order to convert each token into a vector representation. Unlike other deep learning models, BERT has an additional embedding layer which uses Segment embedding and Position embedding. The BERT model further increases the generalization capability of the word vector model to fully represent feature information at the character level, word level, sentence level, and even inter-sentence relationships.

The M-BERT model performs better than BERT in multilingual tasks. The M-BERT model used in this paper is composed of a 12-layer Transformer encoder. In this model, the encoded output of the last layer is taken as text embedding representation to obtain the corresponding vector representation. Bi-LSTM is used to further deepen the text context interaction and capture the local relationship of the text sequence.

### *3.2. Double Attention*

The common attention mechanism mainly uses the position information of sentence sequences to perform its calculations. The double attention mechanism based on this paper passes the input text sequence context information into either the semantic attention layer or the structural attention layer. Different queries will assign different weights to the contents of the source text, so as to capture the potential information. The semantic attention layer focuses on the inherent semantic information of words, while the structuralist layer tends to focus on the correlation between words, through which the text features can be further extracted.

The double attention layer is based on the double attention mechanism to process the context vectors obtained from the encoding layer and then learn the deeper semantic information in the text. The double attention mechanism in this layer aims to obtain the weight matrix R by calculating the correlation between each token in the English and Chinese sequences, respectively, and then weighting and summing the normalized weights and the corresponding key values to obtain the final attention representation. Here, the correlation is reflected by the dot product score matrix, and the main role of dot product attention is to learn self-alignment information, i.e., the interaction information of token pairs. The self-attention mechanism captures the internal relevance of text features by comparing the sequence itself to capture the connection between the sequence and the global features, and its simplified structure is shown in Figure 3.

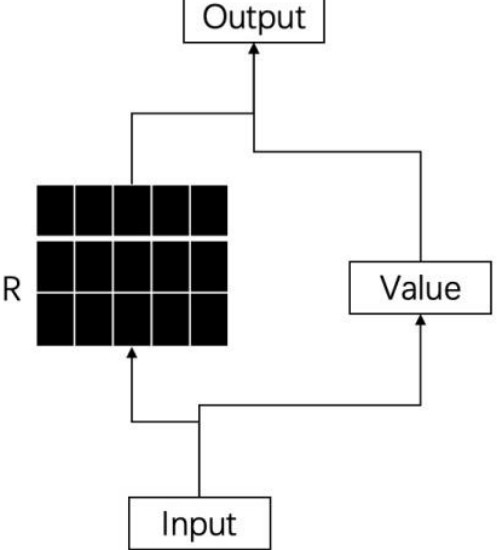

**Figure 3.** Self-Attention Structure.

In order to solve the problem of embedding the context information of the lost word with the word conversion word, this paper introduces the double attention mechanism into the cross-lingual Chinese named entity recognition model. The algorithm makes full use of the features extracted from the pre-trained English named entity recognition model. A double attention mechanism is used to transfer the high-level information of the English model to the Chinese model by drawing on the encoder-decoder attention weight in the Seq2Seq translation model [19]. The attention weight represents the degree of word correlation between the source language and the target language in the model. Due to the equivalence of word correspondence in the two languages, the information extracted from the pre-trained English named entity recognition model can be translated into the Chinese language by using the attention matrix in reverse. Compared with the method based on shared representation, this model uses the pre-trained English named entity recognition model to extract the task-related information and uses the attention matrix to reverse transform, which results in higher transfer efficiency and less noise [20].

*3.3. Adversarial Training Layer*

To better improve the cross-language word vector alignment, this paper constructs a shared semantic space from the source language to the target language. In a recent study, Zhang [21] explores unsupervised methods to learn cross-language word vectors and achieves comparable performance with supervised methods. In this paper, word-level adversarial training is performed to automatically align the word representations of the source and target languages.

To construct the adversarial training, this paper first gives the pre-trained monolingual word vector target language $V_t = \{v_1^t, v_2^t, \ldots, v_N^t\} \in R^{N \times d_t}$ *and source language* $V_s = \{v_1^s, v_2^s, \ldots, v_M^s\} \in R^{M \times d_s}$, where $v_i^t$ and $v_i^s$ are vector representations of words $w_i^t$ and $w_i^s$, respectively; t, s, N, and M indicate dictionary size; and $d_t$, $d_s$ indicate word vector dimensions. Then, a mapping method f is applied to map s to the same semantic space as t:

$$\widetilde{V}_s = f(V_s) = V_s U \tag{2}$$

where $U$ is a transformation matrix, $\widetilde{V}_s$ is the word vector mapped by $s$, and this paper uses singular value decomposition (SVD) to constrain the orthogonality of the transformation matrix $U$ when generating the transformation matrix $U$ in order to reduce the parameter search space:

$$U = AB^T, \ A \sum B^T = SVD(\widetilde{V}_s V_s^T) \tag{3}$$

In order to optimize the mapping method without using additional bilingual information, a multi-layer perceptron $D$ is introduced as a word discriminator in this paper. The role of the word discriminator is to generate a separate tensor by taking the target word vector and the mapping word vector as input features. $D(w_i^*)$ is used to denote the probability that $w_i^*$ comes from $t$. This word discriminator is optimized by minimizing the binary cross-entropy loss:

$$L_{dis}^w = -\frac{1}{I_{t;s}} \cdot \sum_{i=0}^{I_{t;s}} (y_i \cdot log(D(w_i^*)) + (1 - y_i) \cdot log(1 - D(w_i^*))) \tag{4}$$

$$y_i = \delta_i(1 - 2\epsilon) + \epsilon \tag{5}$$

When $w_i^*$ is from the target language word vector, $\delta_i = 1$; otherwise, $\delta_i = 0$. $I_{t;s}$ denotes the number of words sampled together from the vocabulary of t and s, and $\epsilon$ is the smoothed value for adding positive and negative labels. $\Theta_{dis} = \{\theta_D\}$ is the parameter set.

The total loss function consists of two components, a word discriminator $D$ and a mapping method $f$, which optimizes $f$ by flipping word labels and optimally minimizing the total loss function:

$$L_f^w = -\frac{1}{I_{t;s}} \cdot \sum_{i=0}^{I_{t;s}} ((1 - y_i) \cdot log(D(w_i^*)) + (y_i) \cdot log(1 - D(w_i^*))) \tag{6}$$

$$y_i = \delta_i (1 - 2\epsilon) + \epsilon \tag{7}$$

In adversarial learning, the network continuously minimizes the loss of the task discriminator in order to adversarially encourage the shared feature extractor to learn cross-linguistic information for both languages. After training, the task discriminator is unable to distinguish the word vectors of the two languages from the final shared features.

*3.4. Output Layer*

Recurrent neural networks (RNN) usually experience gradient disappearance or gradient explosion during the training process [22]. To solve this problem, long short-term memory networks (LSTM) were born. LSTM can significantly improve the performance of the long-range dependence of the model. The difference between LSTM and the general RNN is that LSTM adds a memory block unit A, and this memory block A includes three parts: the input gate, forget gate, and output gate. The input gate determines how much new information needs to be added to the cell; the forgetting gate is mainly used to control the storage of information in the cell, i.e., to decide what information to discard; and the output gate determines what information is to be output from this cell A.

The Bi-LSTM layer outputs the predicted scores of each label corresponding to each word, and the highest scores can be selected as the labels of the words. However, there are often some invalid label sequences [23]. Therefore, the CRF layer is added to the Bi-LSTM layer, and the CRF layer can obtain the binding rules from the training data, such as: the first word of the sentence starts with B/O, but not with I; in B- label1 and I-label2, label1 and label2 should be of the same type; and O and I-label cannot be combined together. The probability of illegal sequences appearing in the label sequence is greatly reduced, thus improving the accuracy of label prediction.

The two-way LSTM only considers the long-term contextual information of the sentence but does not consider the dependencies between labels, and CRF can ensure the labels are valid by learning the adjacency relationships between labels. In this paper, we use the standard CRF layer on top of the model of named entity recognition to obtain the final sequence annotation.

$$\text{Define prediction score}: \ S(X, y) = \sum_{i=1}^{n} (O_{t,y_t}^{ner} + A_{y_{t-1},y_t}) \tag{8}$$

$$O_t^{ner} = W^{ner} h_t^{ner} \tag{9}$$

where $W^{ner}$ are model parameters, $A_{y_{t-1},y_t}$ is the transfer probability matrix from label $y_{t-1}$ to $y_t$, and n is the length of the input sentences.

For the input $X = \{x_1, x_2, \dots, x_n\}$, the probability of the output best sequence labeled $\hat{y}$ can be defined as

$$p(\hat{y}|X) = \frac{e^{s(X,\hat{y})}}{\sum_{\widetilde{y} \in Y_X} e^{s(X,\widetilde{y})}} \tag{10}$$

where $Y_x$ denotes the set of all possible labels, the numerator s function denotes the score of the correct label, and the denominator s function denotes the sum of the scores of each possible label.

In the CRF model training process, the loss function is defined as:

$$L_{ner} = -\log p(\hat{y}|X) \tag{11}$$

The loss function values are calculated, and the network parameters are continuously updated until the end of the iteration.

## 4. Experimental Results and Analysis

### 4.1. Data Sets

In this paper, we use the Conll2003 data sets for training and the WeiboNER data sets [24] as well as the People's Daily News data sets [25] for experiments. WeiboNER is generated by filtering and filtering the historical data of Sina Weibo from November 2013 to December 2014. It contains 1890 microblog messages and is labeled based on DEFT ERE labeling standard of LDC2014. The data set entity contains four categories: place name, person name, institution name, and administrative name; each category can be subdivided into specific (NAM, such as "Chang SAN" labeled "PER.NAM") and general (NOM, such as "men" labeled "PER.NOM") categories. People's Daily physical data sets is a annotated corpus co-produced by the Institute of Computational Linguistics of Peking University and the Fujitsu Research and Development Center Co., LTD., based on the corpus of People's Daily published in 2004. Both datasets are used for the cross-lingual named entity recognition task, and the People's Daily News data sets is used for the remaining sets of experiments except for the comparison experiments. Both datasets are simplified Chinese datasets, as shown in Table 1.

**Table 1.** NER dataset.

| Dataset | Language | Train | Dev | Test |
|---------|----------|-------|-----|------|
| CoNLL2003 | English | 204,567 | 51,578 (5942) | 46,666 (5648) |
| WeiboNER | Chinese | 1350 | 270 | 270 |
| PeopleDaily2004 | Chinese | 28,046 | 4636 | 4636 |

### 4.2. Experimental Configuration

The experiments in this paper are trained using GPU, the development language is Python, and the deep learning framework is Pytorch. because the model in this paper adds attention layer and Bi-LSTM, which increases the interaction process between sequences, the training speed of the model in this paper is slower compared to the limit model. The experimental parameters are shown in Table 2. BERT embeddings have a dimension of 100.

**Table 2.** Experimental parameters.

| Parameters | Value |
|-----------|-------|
| Epoch | 3 |
| Batch size | 20 |
| Learning rate | $4 \times 10^{-5}$ |
| Dropout | 0.5 |
| Token embedding dimension | 100 |

Batch size should be set according to GPU memory. If it is set to 20, the training test can be conducted quickly, and the learning rate can be adjusted to observe the experimental results. In the local experiment, when the learning rate is $4 \times 10^{-5}$, the experimental results can be optimized.

### 4.3. Experimental Results and Analysis

Currently, the most commonly used evaluation criteria for NER are Precision, Recall, and F1-score.

In order to verify the effectiveness of this paper's model for the cross-lingual named entity recognition task, a comparison experiment of different named entity recognition models is conducted. The experiment was conducted on the same data set using CRF

model, Pipeline model, and M-Bert+Bi-LSTM-CRF model for the cross-lingual NER task, respectively, and the results are listed in Table 3.

**Table 3.** Accuracy of different benchmark models in People-Daily Chinese dataset.

| Model | $p$ | R | F1 |
|---|---|---|---|
| CRF (Peng and Dredze, 2015 [24]) | 0.5698 | 0.2526 | 0.3500 |
| Pipeline Seg.Repr. + NER (Peng and Dredze, 2015) | 0.6422 | 0.3608 | 0.4620 |
| Word2vec + Bi-LSTM-CRF | 0.3199 | 0.5600 | 0.3290 |
| ELMO + Bi-LSTM-CRF | 0.3870 | 0.4428 | 0.4131 |
| M-Bert + Bi-LSTM-CRF | 0.4222 | 0.1843 | 0.4693 |

Compared with the traditional CRF-based model and the traditional Pipeline-based [26] named entity recognition model, the performance of the model in this paper is also improved greatly, indicating the effectiveness of the framework in the task of cross-language named entity recognition. In order to study the influence of different word vector representation methods on the experimental results, the experiment also selects Word2Vec [27], ELMO [28], and BERT, three mainstream word vector representation methods, for a comparative test. Word2Vec is a static word vector representation method considering the context-free information of words. Both ELMO and BERT are dynamic word vector representation methods that can fully represent the contextual semantic and syntactic information of words. The difference is that ELMO can only take into account the unidirectional semantic information of words, while BERT can fully integrate the semantic information of words in both directions of context. The experimental results prove the advantage of using the pre-trained M-BERT [29] model for vector representation in the cross-language named entity recognition model.

From Table 4, this model achieves an F1 value of 53.22% in the People-Daily2004 data set, which is 6.29% better than the baseline model, and an F1 value of 53.71% in the WeiboNER data set, which is 3.39% better than the baseline model, outperforming the other comparison models on both datasets. The experimental results show that the model in this paper has a significant improvement in performance and is able to learn the deep to be information of the text, which effectively improves the cross-language migration model. In addition, compared with Lin et al. (2018) [30], who only apply a shared context encoder to transfer the knowledge, our approach not only includes a language-sharing encoder, but also performs word-level adversarial training to encourage the semantic alignment of words from both languages and a sequence encoder to extract language-agnostic sequential features [31].

**Table 4.** Comparison of the effects of different models on two Chinese datasets.

| Model | WeiboNER (F1) | People-Daily2004 (F1) |
|---|---|---|
| M-Bert + Bi-LSTM-CRF | 0.5032 | 0.4693 |
| M-Bert + Bi-LSTM-CRF + Word-adv + Att | **0.5217** | **0.5208** |
| M-Bert + Bi-LSTM-CRF + Word-adv + Att + xlpos | **0.5192** | **0.5322** |

To investigate the contribution of attention mechanisms to the model, ablation experiments were designed to further analyze the model in this paper. As can be seen from Table 5, the F1 values decreased by 1.8% and 4.8% when the model did not use the attention mechanism. The results show that using the attention mechanism can make full use of the information carried by the input sequence, prevent the model from losing the original1 information with training, and make the model better for cross-linguistic transfer. From the experiments where the location information was removed, it can be observed that the F1 values decreased by 0.85% and 1.08%. It can be seen that the incorporation of location information has some improvement on the model effect. It can be seen from the results of the ablation experiment that better results can be obtained by using the weight of the deep attention layer as the transfer matrix, which also means that the algorithm proposed in this

paper can effectively assist the Chinese named entity recognition by inversely transferring the features extracted from the English model, so that the relevant features extracted from the English pre-trained model can effectively assist the Chinese named entity recognition [32]. Moreover, the attention mechanism [33] integrated with location information can capture deeper semantic dependencies and have more accurate alignment effect, thus improving the recognition effect more obviously.

**Table 5.** Ablation experients.

| Model | WeiboNER (F1) | People-Daily2004 (F1) |
|---|---|---|
| Model | **0.5192** | **0.5322** |
| Without attention | 0.5012 | 0.4834 |
| Without position | 0.5107 | 0.5214 |

*4.4. Case Study*

To illustrate the effectiveness of the proposed model, the first sentence in the test set is used as shown in Figure 4. The previous NER benchmark model M-BERT+BiLSTM+CRF cannot label the word "the boss" as "B-PER I-PER" and classify it as a non-entity, but the model in this paper can correctly label the word "the boss" as "B-PER I-PER" and classify it as a person entity. This indicates that the cross-linguistic NER task is effectively migrated and verifies the effectiveness of adversarial training on the cross-lingual NER task and the positive effect of attention mechanism on the acquisition of boundary information.

**Figure 4.** Case Study.

## 5. Conclusions

In order to improve the accuracy of cross-lingual named entity recognition, this paper proposes a cross-lingual named entity recognition method based on attention mechanism and adversarial training, which firstly introduces the M-BERT pre-trained language model. The obtained word vectors can enhance the word–word dependencies, adds the attention mechanism based on the adversarial migration model and obtains the semantic representation of the aligned word vectors by enhancing the more accurate text representation vectors. Finally, the best entity annotation is obtained. In addition, the accuracy of the cross-lingual NER task is improved by introducing adversarial training to obtain the shared features and word boundary information of both languages. The experimental results show that the method in this paper shows good results for named entity recognition in low-resource

languages; however, name entity recognition not only relies on word-level features, but also on sequential features for entity-type classification. In the future, we will experiment with language-agnostic information at the sentence level.

**Author Contributions:** Methodology, L.H.; Resources, J.D.; Writing—original draft, H.W.; Writing—review & editing, L.Z. All authors have read and agreed to the published version of the manuscript.

**Funding:** This work is supported by R&D Program of Beijing Municipal Education Commission (KM202210009002), the National Natural Science Foundation of China (61972003), and the Beijing Urban Governance Research Center.

**Institutional Review Board Statement:** Not applicable.

**Informed Consent Statement:** Not applicable.

**Data Availability Statement:** Not applicable.

**Acknowledgments:** We would like to thank the anonymous reviewers for their helpful comments. We would like to thank the referees for their comments, which helped improve this paper considerably.

**Conflicts of Interest:** The authors declare no conflict of interest.

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
