# Peer review of "Cross-Lingual Named Entity Recognition Based on Attention and Adversarial Training"

_applsci, doi:10.3390/app13042548_

Round 1
Reviewer 1 Report
Dear Authors,
Well done!
53.22% optimal model (a 6.29% improvement compared to baseline) is a significant improvement to English-Chinese cross-lingual named entity recognition. I work in this area and know the importance of your research.
Kindly help to improve the following areas:
1. Edit the last paragraph of the introduction to make the aim and objectives of this manuscript easier for readers to understand, especially the objectives, so that they can understand better as they read through the paper.
2. The second "Related works" section didn't tell us much about "related works". It would be better if there is a subsection that sheds more light on related works and their shortcomings.....explain the research problem or gap better at the end of this subsection. This section will also reconnect back to the objectives at the end of section 1. Try to better storytellers and let the story flow so that the readers will enjoy your work and get curious look forward to the next section instead of asking questions about what is in the literature (other related works). I know you have some in there, organise them better.
3. Please add links or refs for the datasets used in 4.1. Kindly also describe it better if no previous paper has done that that you can cite.
4. There is a need for a discussion section after the experiment before the conclusion. You have done a fantastic job up to the case study, so don't run out of gas at this point. Tell us how the experimental results have solved the problem or made English-Chinese cross-lingual named entity recognition better by reflecting back to the existing literature and what others have done and what exact improvements your proposed model has achieved. We have this scattered in the experiment section but not too clear. Put them together and tell the story better in this new section.
5. Blow your own trumpet more, you have done a great job, tell us more about the significance of your work in the conclusion. Before the sentences on future research, also tell us the limitations of your proposed model and how future researchers can improve that.
6. Go back to your abstract and modify it to show these corrections.
7. Overall, you need to add more citations/refs. from the introduction to the discussion, especially in the model and experiments sections. Give your readers the opportunity to read as much as possible about the things you are adopting or proposing.
All the best!
Author Response
Distinguished editor,
Thank you for your letter and for the referee's comments concerning our manuscript entitled “Cross-lingual Named Entity Recognition based on Attention and Adversarial Training”. We have studied your comments carefully and have made correction which we hope meet with your approval.
- In the last paragraph of the introduction, we explain the aim and objectives of this paper more clearly. We use the attention mechanism in reverse to improve the efficiency of cross-lingual transfer and reduce the impact of noise in the process of transfer; In order to better align the word vector, we introduced the adversarial training to improve the ability of the model to align between two languages.
- In related work, I added other authors' research on cross-lingual migration, including their proposed methods and emphases, under the subsection "Cross-language migration".
- In 4.1, a brief introduction is made to the data set cited in the paper and we have added links for them.
- We have discussed more experiment before the conclusion.In addition, we added several comparative experiments to explain the reasons for choosing M-BERT.And The effect of adding word-level adversarial training and attention mechanism are explained.
- In the conclusion, we have told reader the limitations of our proposed model and how future researchers can improve that.
- I made some simple changes to the abstract section about data sets.
- We have added more citations/refs from the introduction to the discussion.
Thank you very much.

Reviewer 2 Report
Recommendation: Author should prepare a major revision
The authors proposed a method for cross-lingual entity recognition based on attention mechanism and adversarial training, using resource-rich language annotation data to migrate to low-resource languages for named entity recognition tasks, and outputting changing semantic vectors through the attention mechanism to effectively solve the long-sequence semantic dilution problem.
1. Literature review is incomplete. The authors should provide reviews of machine learning-based models for biological problems, such as the paper with PMIDs: 32004641 and https://doi.org/10.1016/j.knosys.2023.110295.
2. Describe double attention in more detail, and also specify differences from basic attention.
3. The resolution of Figure 2 is too low. Replace it with a high-resolution figure.
4. Detailed information on hyperparameters and how they were tuned should be described so that readers can re-implement the model easily just by reading the paper.
5. The authors should revise English writing carefully and eliminate small errors in the paper to make the paper easier to understand.
Author Response
Distinguished editor,
Thank you for your letter and for the referee's comments concerning our manuscript entitled “Cross-lingual Named Entity Recognition based on Attention and Adversarial Training”. We have studied your comments carefully and have made correction which we hope meet with your approval.
- We reviewed and added more citations/refs from the introduction to the discussion.
- We described double attention in 3.2 and the differences from basic attention.
- We replaced figure 2 with a high-resolution figure.
- In the part of hyperparameter, we have explained the process of parameter adjustment. For example, some parameters need to be adjusted according to the local computer configuration in the process of recurrence. The configuration provided in this paper is only for reference.
- I will use mdpi editing service to make English writing better.
Thank you very much.

Reviewer 3 Report
Accepted with Minor Changes
Abstract: Add 2 lines about study contribution
Introduction: "
As the key semantic information of natural language text, recognition and classifi- | 24 |
cation of named entities is an important element in current natural language processing | 25 |
research. For languages with a large number of speakers and abundant corpus resources, | 26 |
it may be relatively easy to obtain corresponding hand-annotated data, but for most | 27 |
low-resource languages, there is no data of sufficient size for training deep neural net- | 28 |
work named entity recognition models, which poses a high demand for cross-lingual | 29 |
migration tasks |
" This start is not suitable, authors should start with introducing thier research on the basis of state-of-art and limitations of previous researchs and need for your study
Related Work: This section is articulated well but authors are advised to give simplist and understand able definations of variable in start.
Model need more explaination
Explain Table 2.
Conclusion: Authors are advised to give more attention to this section and extend it with contribution to theory and practice.
I appriciate the efforts of authors for such a nice work.
After adding my comments It is recommended for publication
Author Response
Distinguished editor,
Thank you for your letter and for the referee's comments concerning our manuscript entitled “Cross-lingual Named Entity Recognition based on Attention and Adversarial Training”. We have studied your comments carefully and have made correction which we hope meet with your approval.
- Introduction: We modify the introduction with introducing the reseaerch on the basis of state-of-art and limitations of previous researchs.And then we introduce our study and the importance of our study.
- Related Work:In the part of hyperparameter, we have explained the process of parameter adjustment. For example, some parameters need to be adjusted according to the local computer configuration in the process of recurrence. The configuration provided in this paper is only for reference.
- In the conclusion, we have told reader the limitations of our proposed model and how future researchers can improve that.
The function of parts of the model was more clearly divided and subsequently interpreted by parts.
Thank you very much.

Round 2
Reviewer 2 Report
The authors did not respond to the comments (1,2,3) and ignored it. I have no choice but to give reject.
Author Response
Distinguished editor,
I am sorry that I did not pay enough attention to your first revision and misunderstood the questions you raised. This time, I have made careful modifications according to your suggestions 1, 2 and 3.
- I added "miRNA-Disease Association Prediction with Collaborative Matrix Factorization" as the 15th citation. In addition, the relationship between biological attention mechanism and nlp attention mechanism is introduced in 2.3.
- In 3.2, we further explain the double attention layer and explain its difference from ordinary attention.
- We redrew the model diagram and put it into the manuscript. I'm sorry that we didn't correct the corresponding problem due to our previous negligence.
Thank you very much.

Round 3
Reviewer 2 Report
STILL, authors did not respond to comment 1.
Author Response
Distinguished editor,
I am very sorry that I made an oversight in the process of adding references, and now I will correct it. I added "miRNA-Disease Association Prediction with Collaborative Matrix Factorization" as the 15th citation.
Thank you very much.

Round 4
Reviewer 2 Report
The reviewer's response is too insincere. I don't know how many times I've said the same thing. Two papers for comment 1 were presented from the beginning, but one was mentioned in the second request. This is the 3rd request. Is it disregard for reviewer?